# An Ultra Low Power Integer-N PLL with a High-Gain Sampling Phase Detector for IOT Applications in 65 nm CMOS

Javad Tavakoli [1], Hossein Miri Lavasani [2,*] and Samad Sheikhaei [1]

1. School of Electrical and Computer Engineering, College of Engineering, University of Tehran, North Kargar St., Tehran P.O. Box 14395-515, Iran; javad_tavakoli@ut.ac.ir (J.T.); sheikhaei@ut.ac.ir (S.S.)
2. Department of Electrical, Computer, and Systems Engineering, Case Western Reserve University, Cleveland, OH 44106, USA
* Correspondence: sxm1243@case.edu; Tel.: +1-216-368-4436

**Abstract:** A low-power and low-jitter 1.2 GHz Integer-N PLL (INPLL) is designed in a 65 nm standard CMOS process. A novel high-gain sampling phase detector (PD), which takes advantage of a transconductance (Gm) cell to boost the gain, is developed to increase the phase detection gain by ~100× compared to the Phase-Frequency Detectors (PFDs) used in conventional PLLs. Using this high detection gain, the noise contribution of the PFD and Charge Pump (CP), reference clock, and dividers on the PLL output is minimized, enabling low output jitter at low power, even when using low-frequency reference clocks. To provide a sufficient frequency locking range, an auxiliary frequency-locked loop (AFLL) is embedded within the INPLL. An integrated Lock Detector (LD) helps detect the INPLL locked state and disables the AFLL to save on power consumption and minimize its impact on the INPLL jitter. The proposed INPLL layout measures 700 μm × 350 μm, consumes 350 μW, and exhibits an integrated phase noise (IPN) of −37 dBc (from 10 kHz to 10 MHz), equivalent to 2.9 ps rms jitter, while keeping the spur level 64 dBc lower, resulting in jitter figure of Merit ($FoM_{jitter}$) ~−236 dB.

**Keywords:** phase-locked loop; PLL; phase detector; PD; sampling PD; INPLL; lock detector

## 1. Introduction

The advent of low-power, high data-rate data communications has paved the way for an array of novel Internet of Things (IoT) applications, ranging from smart homes to remote medical diagnostics and environmental sensing. Meeting the escalating data rate demands necessitates stringent phase noise (PN) and spur specifications for the local oscillator (LO). This is critical as the LO's phase noise profoundly influences the error vector magnitude (EVM) floor in wireless transceivers, particularly after calibrating for other signal impairments. These exacting constraints on PN and jitter have introduced considerable complexity to the design of phase-locked loops (PLLs) employed in LOs.

Traditional PLL architectures featuring a conventional Phase-Frequency Detector (PFD), combined with a Charge Pump (CP) for phase detection and Voltage-Controlled Oscillator (VCO) tuning, have suffered from inadequate jitter performance. This limitation stems from the inherently low phase detection gain in the PFD and the limited switching speed of transistors, especially PMOS devices, utilized in the CP. To address this issue, numerous design alternatives have been proposed in the existing literature [1–20]. A prevalent approach among low-jitter PLL designs in the literature involves using a subsampling structure [1–3,14–17]. This innovation obviates the necessity for feedback frequency dividers, PFDs, and CPs, effectively eliminating their phase noise contributions and reducing overall PLL jitter. Furthermore, by directly capturing the swift transitions produced by the VCO, subsampling achieves a significantly higher phase detector gain, thereby minimizing the phase noise contribution of the phase detector (PD) to the PLL's overall phase noise [1].

Despite their capacity to achieve low phase noise (and correspondingly low jitter), subsampling designs tend to compromise on spur levels to achieve improved phase noise performance. Typically, addressing the issue of spurs leads to a significant increase in power consumption. One approach to balance this trade-off is the utilization of a reference sampling PLL (RS-PLL) [4]. The RS-PLL assesses the VCO's phase error by sampling the reference sine wave with a VCO square wave, significantly enhancing spur and noise performance without adversely impacting jitter or power efficiency. Nevertheless, due to the exponential nature of the RC time constant, the phase detection window remains relatively small, resulting in a limited acquisition range. Other researchers have explored methods like double sampling and retiming techniques to mitigate phase noise [5], although these approaches inherit some of the issues associated with subsampling PLLs. In the context of higher frequency PLLs, another technique, injection locking, has been applied to diminish phase noise [6,19]. In this technique, a copy of the reference signal is injected into the VCO to suppress VCO phase noise. However, this strategy introduces periodic disturbances in the VCO, yielding reference spurs of relatively significant magnitude [6].

In pursuit of a low-jitter solution with minimal impact on power consumption, we introduce a novel INPLL structure centered around a high-gain linear sampling Phase Detector. By employing this innovative high-gain sampling PD, we achieve a substantial increase in the phase detection gain more than 100 times faster than traditional PLLs. Furthermore, the practical Detectable Phase Error (DPE) exhibits remarkable enhancement, scaling up by as much as tenfold. The gain of our proposed PD is adaptive, providing the flexibility to adjust it by modulating the current in the current source. To expand the frequency locking range, we have incorporated an Auxiliary Frequency-Locked Loop (AFLL) into the design. Additionally, we have integrated a lock detector circuit to discern the locked state and, when required, disable the AFLL. This strategic move minimizes the AFLL's influence on INPLL output jitter and contributes to an overall reduction in power consumption.

Regarding paper structure, our content unfolds as follows: in Section 2, we offer an insight into the conventional phase detection approach, while shedding light on its limitations. This section also includes a succinct examination of the various sources contributing to the overall output noise in PLLs. Section 3 outlines our proposed PLL architecture, detailing the innovative high-gain Linear Sampling Phase Detector (LSPD). Section 4 is dedicated to presenting the results of simulations conducted on the PLL, comparing its performance to state-of-the-art PLLs operating within the same frequency range. Finally, Section 5 provides the concluding remarks.

## 2. Exploring a 1.2 GHz INPLL Design with High-Gain LSPD

### 2.1. Constraints of Traditional PFD-CP PLL

Figure 1a illustrates a basic three-state PFD and CP structure. Additionally, the linear transfer characteristic of this structure is shown, where I is the dc value of the charge pump current, $\Delta\Phi$ is the phase error at the PFD input, and $\overline{i_{cp}}$ is the average output current. The PFD in this configuration employs two resettable D Flip-Flops (DFFs) and an AND gate. Both D inputs of the DFFs are connected to 1. The PFD analyzes the reference and the divided VCO signal (REF and FB), producing 'up' and 'down', signals based on their phase relationship. Expressly, when REF leads FB, UP is set to 1; conversely, when FB leads REF, DOWN is set to 1. To address the 'dead zone' challenge within the PFD, the reset path's delay is extended by introducing delay blocks, like inverter buffers. As a result, the PFD can produce 'up' and 'down' pulses even when there's no phase difference. The duration of these pulses corresponds to the reset path's propagation delay ($\tau_{\text{PFD}}$), during which the CP introduces noise in each reference cycle.

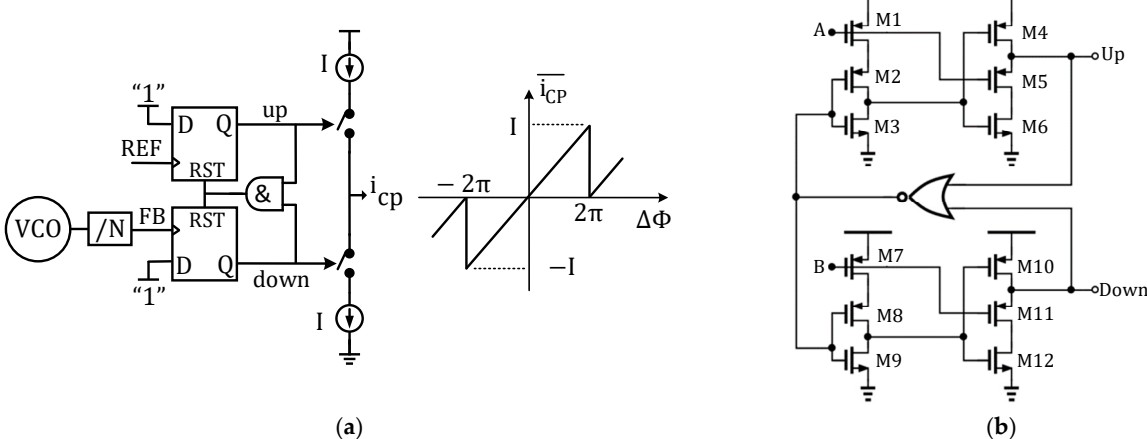

(a)    (b)

**Figure 1.** (**a**) Structure of a conventional three-state PFD and CP and its linear transfer characteristic; (**b**) TSPC PFD structure.

Another traditional structure employed for phase detection is the True Single-Phase Clocking (TSPC)-based PFD, as depicted in Figure 1b. This configuration is preferred for its reduced noise contribution [21].

Despite their simplicity, both designs encounter notable phase errors primarily attributable to the inherent switching speed of PMOS devices within the Charge Pump. To put this into context, consider that a typical PFD-CP configuration designed for sub-100 nm standard CMOS processes encounters challenges when detecting time differences less than 20 ps [22]. Furthermore, conventional PFD-based designs contend with low PFD detection gain, typically around $V_{DD}/2\pi$, which significantly influences PLL locking and acquisition. Consequently, these designs are often less suitable for low-power applications [21].

*2.2. Noise Contributions in Conventional PFD-CP PLL*

Figure 2a illustrates the architecture of the conventional INPLL, showcasing key components such as the PFD, CP, Loop Filter (LF), VCO, and ÷N Divider. To streamline the design, we employ a basic 2nd-order loop filter composed of a series connection of R1 and C1, operating in parallel with C2.

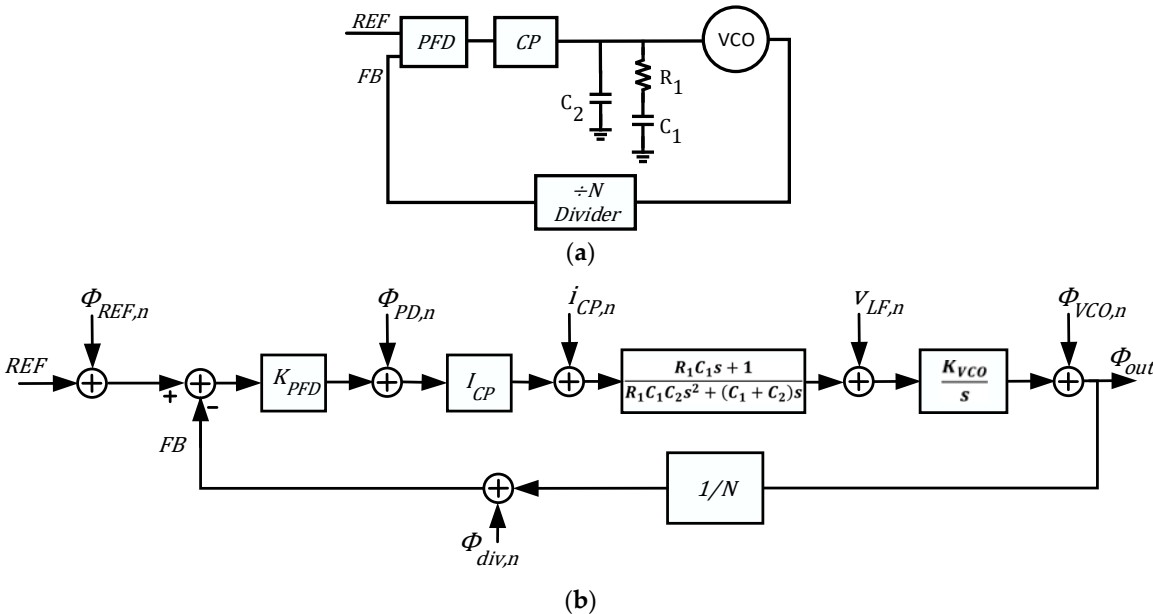

**Figure 2.** (**a**) Architecture of the conventional INPLL; (**b**) linear phase domain model of conventional PFD-CP PLL.

Figure 2b presents a linear phase domain model of this conventional INPLL, capturing the inherent noise sources within the system. This figure highlights the sources of noise: reference path phase noise, $\Phi_{REF,n}$; PFD phase noise, $\Phi_{PD,n}$; CP noise current, $i_{CP,n}$; the loop filter voltage noise, $V_{LF,n}$; VCO phase noise, $\Phi_{VCO,n}$; divider phase noise, $\Phi_{div,n}$. Additionally, $\Phi_{out}$ stands for the PLL output phase.

In order to comprehend the impact of various noise sources on the PLL, it is essential to determine the noise transfer functions (NTF) for each source. The NTFs for the reference signal (REF), Phase-Frequency Detector (PFD), Charge Pump (CP), Low-Pass Filter (LPF), and the ÷N divider can be expressed as follows:

$$\text{NTF}_{\text{REF}} = \text{NTF}_{\text{DIV}} = \frac{N \cdot LG(s)}{1 + LG(s)} \tag{1}$$

$$\text{NTF}_{\text{CP}} = \frac{1}{K_{\text{PFD}} \cdot I_{\text{CP}}} \frac{N \cdot LG(s)}{1 + LG(s)} \tag{2}$$

$$\text{NTF}_{\text{PFD}} = \frac{1}{K_{\text{PFD}}} \frac{N \cdot LG(s)}{1 + LG(s)} \tag{3}$$

$$\text{NTF}_{\text{VCO}} = \frac{1}{1 + LG(s)} \tag{4}$$

$$\text{NTF}_{\text{LPF}} = \frac{C_1}{C_1 + C_2} \cdot \frac{K_{\text{VCO}}}{\left(1 + \frac{s}{\omega_p}\right)} \cdot \frac{1}{1 + LG(s)} \tag{5}$$

where N represents the division modulus, $K_{PFD}$ signifies the PFD phase detection gain, $I_{CP}$ denotes the Charge Pump current, and $K_{VCO}$ indicates the VCO voltage-to-frequency conversion gain. LG(s) represents the loop gain of the PLL, where "s" is the complex frequency parameter. In other words, the transfer function LG(s) typically involves the Laplace-transformed representation of the loop gain of the system. Here, "s" is a complex number denoted as s = σ + jω, with σ being the real part and ω being the imaginary part. The expression for LG(s) is as follows:

$$\text{Loop Gain} = \text{LG(s)} = \frac{I_{\text{CP}} \cdot K_{\text{PFD}}}{N} \frac{K_{\text{VCO}}}{s^2(C_1 + C_2)} \frac{1 + s/\omega_z}{1 + s/\omega_p}, \omega_z = \frac{1}{RC_1}, \omega_p = \omega_z\left(\frac{C_1}{C_2} + 1\right) \tag{6}$$

Upon examination of Equations (1)–(6), it becomes evident that an increase in the phase detector's gain leads to a reduction in noise from both the PFD and CP. Furthermore, as LG(s) rises with $K_{PFD}$, the noise contribution of VCO and LPF diminishes, providing additional motivation for increasing the PFD gain.

## 3. Design of the Proposed PLL Utilizing LSPD

### 3.1. Proposed High-Gain LSPD

To enhance the PFD gain and reduce the minimum DPE, a transition is made from the PFD's digital output to an analog output approach. This transformation involves the introduction of a sampling capacitor, constantly charged by a fixed current source, leading to the creation of a linear ramp voltage across the capacitor, as depicted in Figure 3a. As this voltage ascends from the ground (GND) to the supply voltage (VDD), it is consistently sampled at every edge corresponding to the reference (REF) and feedback (FB) signals. This method effectively translates the phase error between the REF and FB signals into a voltage difference, denoted as ΔV. This voltage difference is subsequently converted into a current using a transconductance (Gm) stage, which is then introduced into the loop filter

to generate the necessary voltage adjustments for VCO tuning. The resulting gain of the Phase Detector can be mathematically expressed as:

$$K_{PD} = \frac{\Delta V}{\Delta \phi} = \frac{\Delta V}{\Delta t} \cdot \frac{\Delta t}{\Delta \Phi} = \frac{\frac{I}{C}}{2\pi f_{REF}} \tag{7}$$

where $f_{REF}$ represents the frequency of the input reference clock. Assuming a constant $f_{REF}$, the PD's gain is determined by $\frac{I}{C}$, which is equivalent to the slope of the ramp voltage. The maximum DPE is directly proportional to this term, as illustrated in Figure 3b. As previously mentioned, the proposed PD necessitates both REF and FB edges to occur during the rising phase from GND to VDD, which can be referred to as the "Phase Detection Window" (PDW). This time window dictates the maximum timing error or phase error that the PD can detect. By implementing a Lock Detector block, this timing window serves to identify out-of-lock conditions within the PLL.

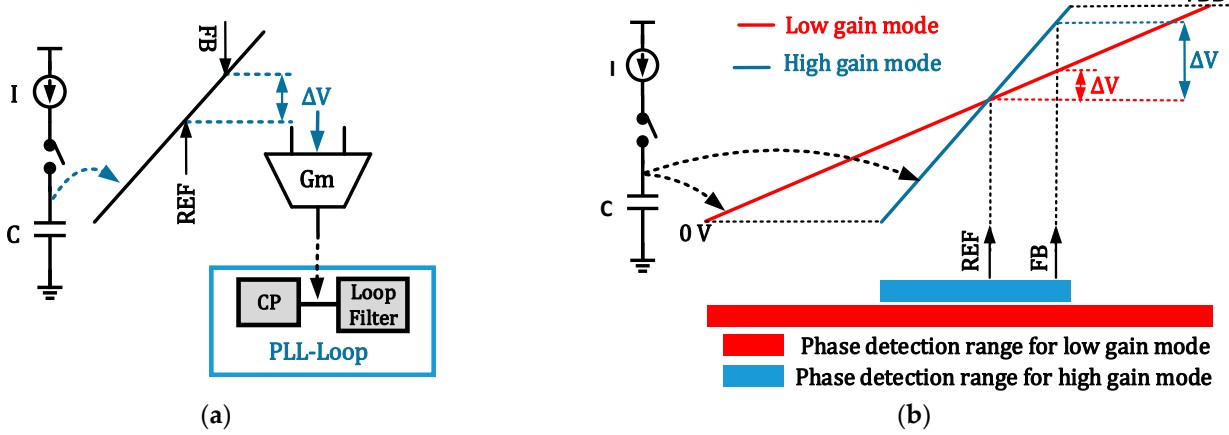

(**a**)  (**b**)

**Figure 3.** (**a**) Conceptual structure of the proposed LSPD (translating phase error into a voltage difference and then converting it into current injected into the loop filter); (**b**) trade-off between phase detection gain and range.

Figure 3b visually demonstrates a trade-off between the detection gain and the phase detection range. Higher gains (indicated by the blue ramp voltage with a steeper slope) correspond to a smaller phase detection window (depicted as a blue rectangle). The phase detection gain can be adjusted according to the specific design requirements by altering the current "I" supplied by the current source. With the constant slope of the voltage ramp, the proposed PD consistently maintains a linear phase detection gain for all phase errors within the phase detection range. Consequently, it is categorized as a linear PD. For this design, a current value of I = 500 µA and a capacitance value of C = 100 fF have been selected, resulting in a slope of approximately 5 GV/s and a corresponding phase detection range of approximately 200 ps.

Knowing that current injection occurs solely within a portion of $T_{REF}$, which we denote as the Gm window, the gain of the cascaded PD-Gm configuration can be expressed as follows:

$$K_{PD-Gm} = K_{PD} \times I_{Gm} \times \alpha \tag{8}$$

where $K_{PD}$ is the PD gain, $I_{Gm}$ is the injected current, and $\alpha$ is the ratio of the injected time (i.e., Gm window) to the $T_{REF}$, i.e., $\alpha = \frac{T_{inj}}{T_{REF}}$. As an example, for slope = 5 GV/s, $f_{REF}$ = 1 MHz and $\alpha$ = 0.02, $K_{PD-Gm} = \frac{5000}{2\pi} \times I_{Gm} \times 0.02 = \frac{100}{2\pi} \times I_{Gm}$, which is two orders of magnitude larger than that of a conventional PD with a similar current. Additionally, accordingly to this high gain, the $G_m$ noise is largely suppressed when referred to the input. Thus, the $G_m$ block can be biased with small bias currents (<10 µA).

Due to the considerable value of $K_{PD}$, approximately 5 GV/s, even a minor 1 ps timing error yields $\Delta V$ of about 5 mV. This value significantly surpasses the typical input

offset voltage of a Gm cell, leading to a notable enhancement in the minimum DPE for the proposed LSPD. A more comprehensive schematic of the proposed LSPD, featuring the Gm block and corresponding timing diagrams, is presented in Figure 4.

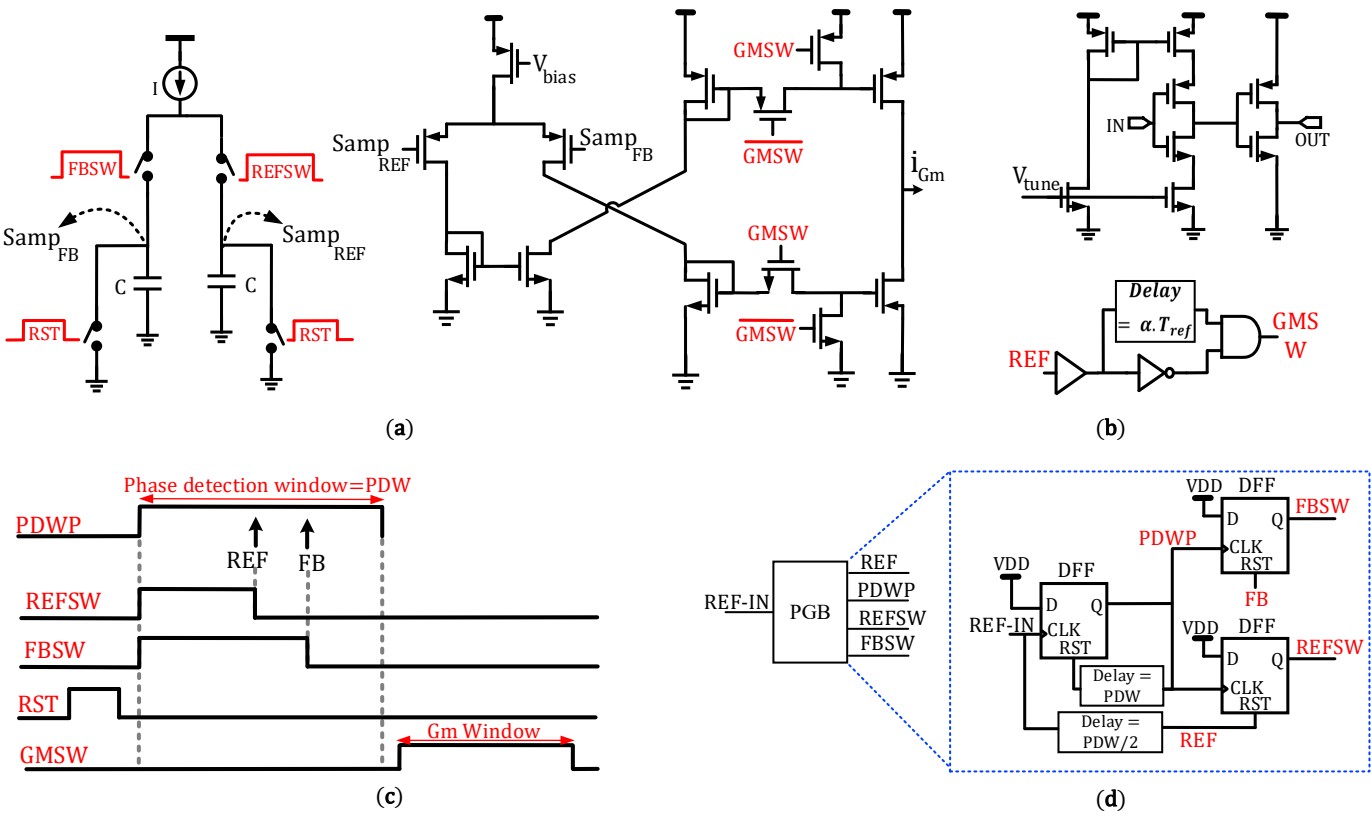

**Figure 4.** (**a**) Schematic of the proposed LSPD-Gm; (**b**) tunable delay cell and GMSW pulse generation circuit; (**c**) timing diagram of LSPD pulses; (**d**) Pulse Generator Block (PGB); (REF-IN: Input Reference, PDWP: Phase Detection Window Pulse, REF: input to the loop, GMSW: $G_m$ window pulse, REFSW&FBSW: pulses determining the charging interval of the sampling capacitors).

To initiate the phase detection process, the consideration of a detection window is essential. This window, termed the "phase detection window" (PDW), is established using the rising edge of the input reference clock (REF-IN) and a predefined delay, as illustrated in Figure 4c,d. As indicated in Figure 3, REF and FB are the two inputs to the PD. From these inputs, two signals, REFSW and FBSW, are derived to facilitate the detection process. REFSW follows the rising edge of the PDW and resets in synchrony with the rising edge of the REF signal. Similarly, FBSW follows the rising edge of the PDW and resets with the rising edge of the FB signal (Figure 4c). When REFSW and FBSW are set, the switches are closed, permitting the current to charge the capacitors (Figure 4a). Once these signals reset, the switches disconnect, and the voltage across the capacitors is employed to interface with the Gm stage to generate the necessary current.

The PDW Pulse (PDWP) is generated utilizing an edge-to-pulse circuit, as depicted in Figure 4d. The tunable delay cell is realized through a straightforward topology detailed in [1] (Figure 4b).

During each clock signal cycle, both capacitors are reset utilizing two switches, as depicted in Figure 4a. Following this reset, the current source I commences charging these capacitors upon the arrival of the positive edge of the PDWP signal. PDWP is synchronized with REFSW and FBSW, which are the pulses responsible for regulating the current through each sampling capacitor. These pulses are generated using the Pulse Generator Block (PGB), as illustrated in Figure 4d. The charging process is interrupted by the positive edge in the corresponding REF or FB signal.

Once this process is concluded, the differential voltage between the two sampled voltages, Samp$_{REF}$ and Samp$_{FB}$, correlates with the timing error between the REF and FB signals at the input of the LSPD. These sampled voltages are subsequently applied to the G$_m$ block, which produces the current flowing into the loop filter. It is crucial to choose the differential pair sizing and bias current such that the input offset voltage remains well below the desired minimum value of $\Delta V$. The resultant current, denoted as $I_{Gm}$, is then introduced into the loop filter using a gate-switching structure, activated when the GMSW pulse is in its high state. Indeed, the Gm circuit in Figure 4a incorporates a conventional current mirror augmented by a gate-switching technique. The GMSW signal is generated using the circuit shown in Figure 4b.

In Figure 5a, the transient simulation results of the proposed phase detector are presented alongside the timing diagram for the RST, REFSW, and FBSW signals. For this simulation, an input time difference of 10 ps is assumed, resulting in a voltage difference of approximately 50 mV between the two sampled voltages, Samp$_{REF}$ and Samp$_{FB}$. This voltage difference determines the phase detection gain as follows:

$$PD \ gain = \frac{\Delta V}{\Delta \Phi} = \frac{\Delta V}{\Delta t} \cdot \frac{\Delta t}{\Delta \Phi} = \frac{50 \ mV}{10 \ ps} \times \frac{1}{2\pi T_{REF}} = \frac{5000}{2\pi}, \ F_{REF} = 1 \ MHz \qquad (9)$$

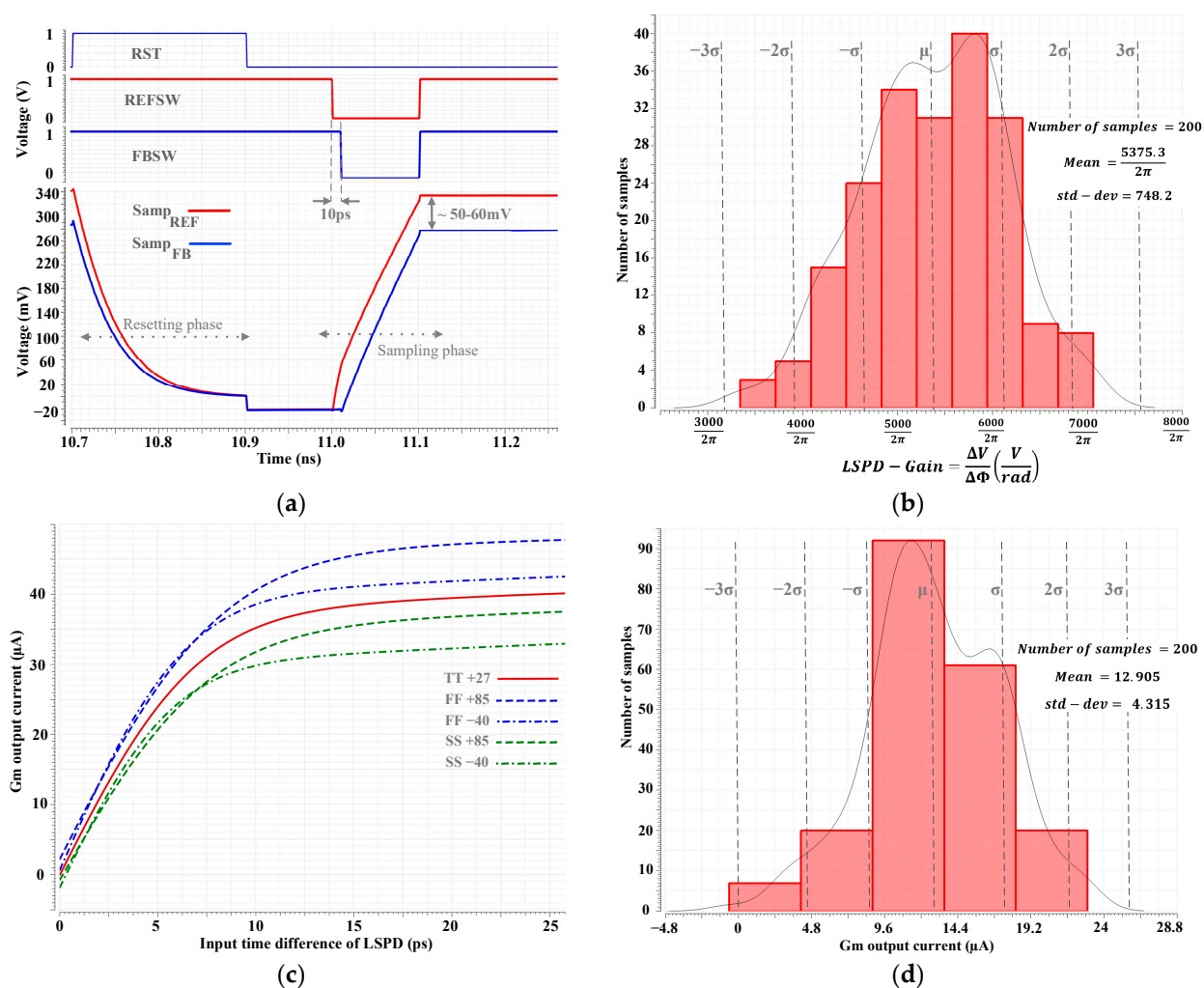

**Figure 5.** Simulated performance of the proposed LSPD showing: (**a**) transient response during the phase detection; (**b**) Monte Carlo results of the phase detection gain; (**c**) Gm output current across process corners and temperature; (**d**) Monte Carlo results of the Gm output current for various input timing errors.

In Figure 5b, the Monte Carlo simulation results illustrate the phase detection gain in the presence of process mismatch for an input time difference of 10 ps. These results showcase a relatively minor standard deviation, underscoring the design's resilience in the presence of the process mismatch. The simulated response of the cascaded LSPD-Gm block to input timing errors is presented in Figure 5c. Here, the Gm output current is depicted for various input timing differences, considering multiple process corners and temperature variations. The Gm input differential pair is biased at I = 40 μA. Furthermore, Figure 5d displays the Monte Carlo simulation results of the Gm output current in the presence of process mismatch for an input timing difference of 2.5 ps. The data portrays a similarly modest standard deviation, affirming the robust performance of the Gm cell in the presence of process mismatch.

### 3.2. Proposed Sampling Integer-N PLL

The proposed integer-N PLL is depicted in Figure 6a. The cascaded LSPD-Gm block fulfills the role of phase detection and current injection to create $V_{ctrl}$ for the Voltage-Controlled Oscillator (VCO). To shape the loop filter, values of R = 80 kΩ, C1 = 100 pF, and C2 = 10 pF have been chosen. The Pulse Generator Block (PGB) is tasked with supplying the REF, PDWP, REFSW, and FBSW signals, which are indispensable for the operation of the LSPD, as indicated in Figure 4c,d. The PLL is designed for IoT application within the 2.4 GHz industrial, scientific, and medical (ISM) band, with specific emphasis on Bluetooth Low Energy (BLE) compatibility. Consequently, the output frequency is established at 1.2 GHz to prevent frequency-pulling issues. As needed, an external frequency doubler positioned outside the loop will up-convert the output to the 2.4 GHz band. In line with this frequency plan, the reference frequency is designated as 1 MHz, effectively encompassing all BLE channels.

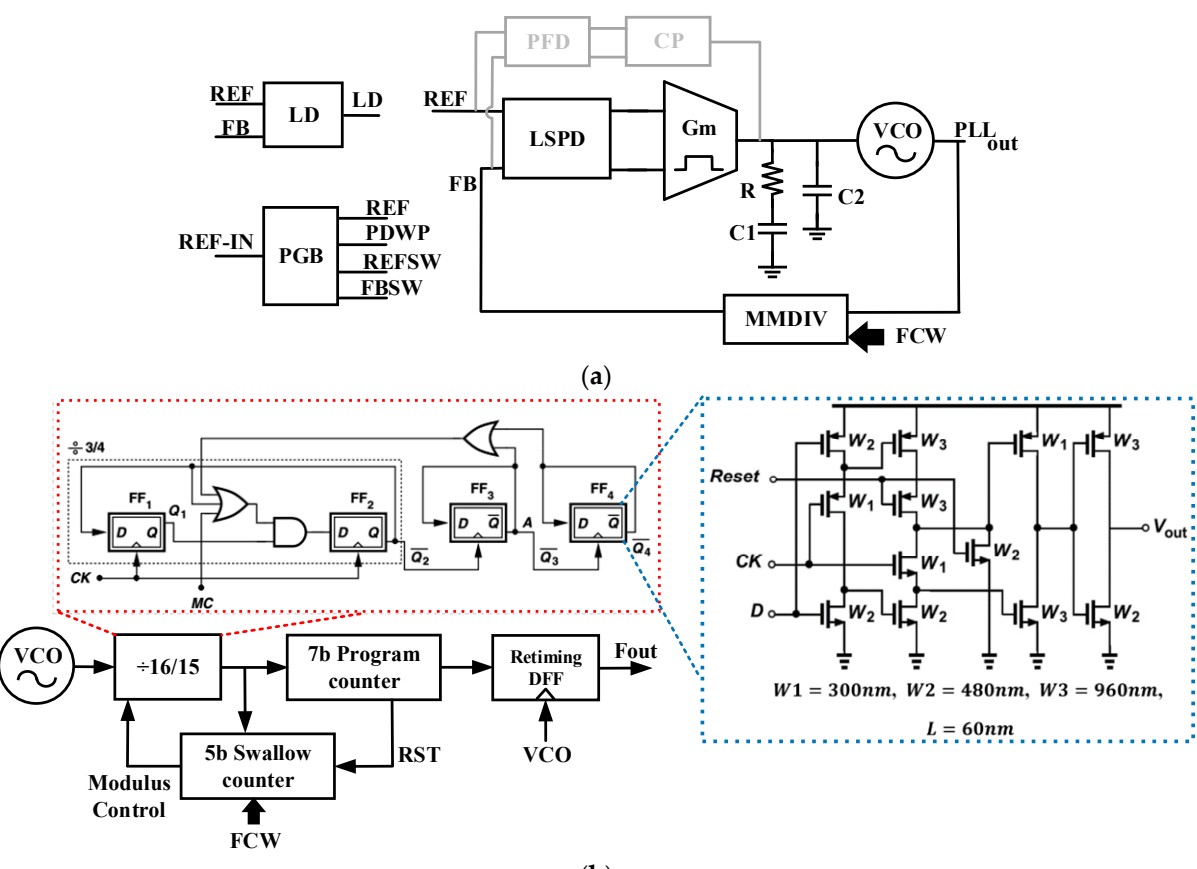

**Figure 6.** (**a**) Block diagram of the proposed INPLL (details of PGB and LD are explained in Figure 4d and LD's section respectively); (**b**) utilized MMD structure.

In the feedback path (Figure 6b), a multi-modulus divider is incorporated, employing the swallow-counter-based structure [23]. This Multi-Modulus Divider (MMD) comprises a dual-modulus prescaler with a divide-by-15/16 capability, a 7-bit program counter, and a 5-bit swallow counter. The schematic for the divide-by-15/16 block and the TSPC-DFF schematic utilized in the structure are also shown in Figure 6b. The utilization of this MMD equips the proposed PLL to encompass all BLE channels effectively. For the divide-by-15/16 prescaler, True Single-Phase Clock (TSPC) DFFs are employed, while standard logic is applied in other dividers. It is notable that asynchronous dividers often exhibit issues related to the propagation of timing jitter within DFFs. To mitigate this concern, a synchronizer DFF is positioned as the last DFF in Figure 6b [24]. This synchronizer DFF ensures that the rising edge of the last DFF synchronizes with the rising (or falling) edge of the VCO output signal, reducing jitter.

To guarantee the correct PLL startup and accelerate the frequency acquisition process, an auxiliary standard PFD-CP based loop is used. Lacking the frequency detection capability, the proposed LSPD has a limited frequency locking range similar to any other phase detector. The auxiliary loop is only used during the PLL startup transient to accelerate locking to the desired frequency. Once this process is completed, the auxiliary loop is turned off to save power. The PLL locked state is determined using a lock detector block, which will be explained in the next section. The conventional PFD also uses a TSPC-based architecture to reduce power consumption (Figure 1b). The CP will also use a simple current steering topology.

### 3.3. Lock Detector (LD) Block

The primary tasks of the LD are (i) to detect the locked state of the PLL and alert the loop to switch off the auxiliary PFD-CP and (ii) to detect when the PLL goes out-of-lock and, consequently, turn on the auxiliary PFD-CP. The conceptual block diagram of the LD is depicted in Figure 7a. In the first stage, two counters, CNT1 and CNT2, are always counting with the positive edges of the REF and FB pulses. Both are $N_1$-bit counters. Whenever each counter reaches the value A, it resets both counters and starts counting again from zero. The outputs of these counters are given to an equality comparator; if the output values of the counters are equal, the output of the comparator will be one. Since the frequency difference between REF and FB pulses is usually large at the startup, the two counters of the first stage always exhibit significantly different values. However, as the PLL approaches the phase-locked state, the counters' output values get closer to each other, ultimately reaching the same value and indicating the phase-lock state.

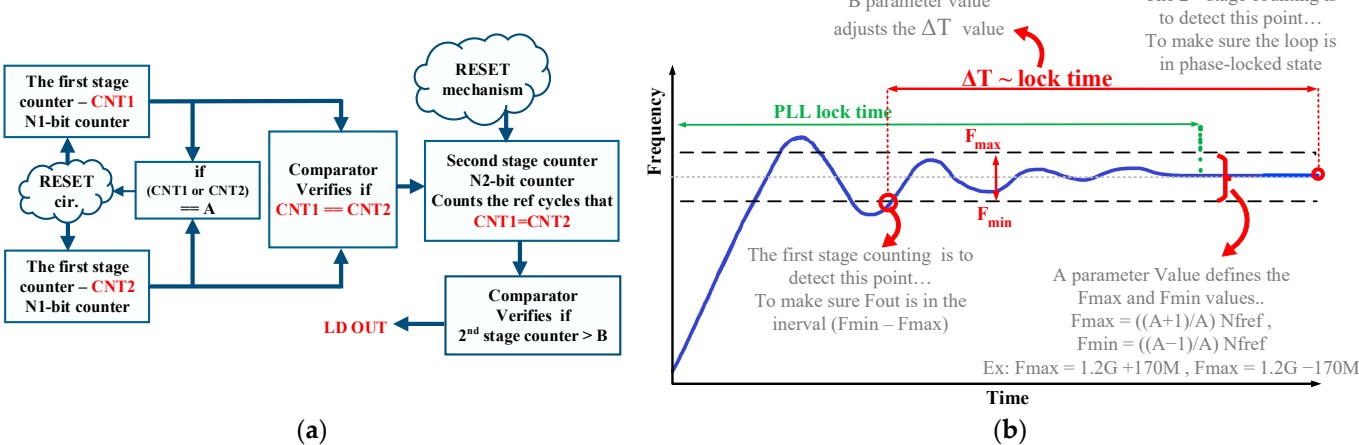

**Figure 7.** (**a**) Conceptual block diagram showing the LD operation; (**b**) lock detection mechanism.

In the second stage, the $N_2$-bit counter counts the number of cycles where the (CNT1==CNT2) condition is satisfied. Another equality comparator in the second stage

verifies whether a certain number of cycles (identified as "B") are counted, after which the LD is set. The phase-lock state is detected by the second-stage comparator, asserting the output signal of the LD block. Reset mechanisms are embedded in the LD block, allowing the structure to quickly recognize the transition of the PLL loop from the phase-locked to the out-of-lock state and reset the LD. The main parameters affecting the operation of the LD block are A, $N_1$, B, and $N_2$. In the proposed design, these parameters are set as $A = 7$, $N_1 = 3$, $B = 35$, $N_2 = 6$. The essential operation of the LD block at the PLL startup and moving into the phase-locked state is shown in Figure 7b. The frequency starts to increase and reaches its final value after PLL lock-time. Depending on the PLL transfer function, some overshooting behavior may be observed in the response, which can be minimized by adjusting the phase margin as necessary [23].

To find the frequency error, $f_{ref}$–$f_{fb}$, the condition in which the two counters of the first stage produce an equal output for A cycles should be investigated. In such cases, the maximum timing error $\Delta T = |T_{ref} - T_{fb}|$ is $\sim \frac{1}{A} T_{ref}$ (if $T_{ref} < T_{fb}$). In other words, the frequency error is in the range $\frac{A-1}{A} \frac{F_{out}}{N} < |f_{ref} - f_{fb}| < \frac{A+1}{A} \frac{F_{out}}{N}$ (N is the division modulus). For example, for A = 7 and $F_{out} = 1.2$ GHz, the frequency range would be $1.03$ GHz $< f_{out} < 1.37$ GHz. In transient state (i.e., PLL is not locked), the PLL frequency moves towards the desired frequency, and frequency error decreases. When the PLL output frequency reaches the frequency interval specified above (shown as $F_{min} < f < F_{max}$), the output of the first-stage comparator remains one. Then, the LD block works by first detecting when the frequency gets to the specified range (first stage). Once this point is reached, the counters in the second stage create a time delay equal to the expected locking time of the PLL. LD detects that PLL will lock if the frequency remains in the specified range (Figure 7b).

Figure 8 shows the block diagram of the LD. The reset mechanism described above is comprised of three separate paths. These three paths are shown in blue (Figure 8). The first path is activated when the counters of the first stage have an unequal value. The second path produces a reset signal if the FB edge arrives outside the PDW. In this case, the PLL is no longer able to correct for the phase error using the LSPD. As such, the auxiliary PFD-CP is engaged to correct the phase error and achieve phase locking. The third path is involved when the PLL is in phase-locked mode while switching to a different frequency channel is desired. In this case, the PLL is released from the phase-locked mode, and the auxiliary PFD-CP is enabled to lock the PLL to the new frequency. To detect changes in the multi-bit input that determines the frequency, i.e., Frequency Control Word (FCW), each bit is compared with the delayed version of the same bit. A pulse with a certain width is produced upon detecting a change in the bit. This simple structure is shown in the upper part of Figure 8. The generated pulse width (delay applied to each FCW bit) should be long enough to reset the counter in the second stage.

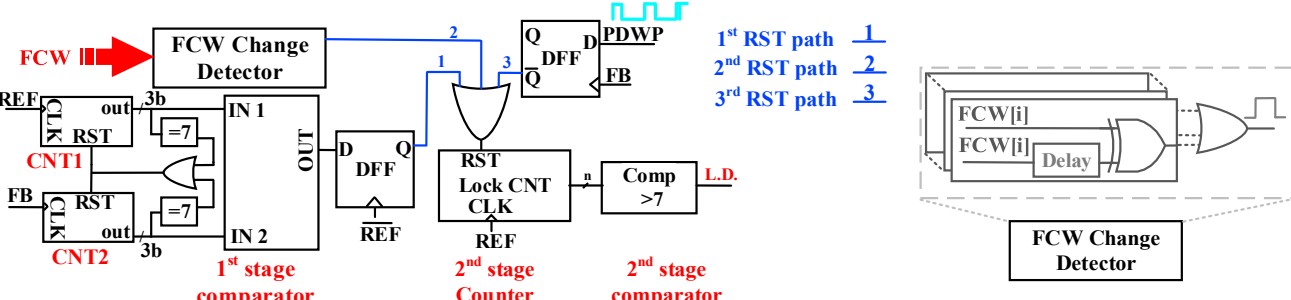

**Figure 8.** Block diagram of the LD showing both stages and the reset paths (FCW: Frequency Control Word).

The simulated performance of the proposed lock detector is illustrated in Figure 9. As observed, during the transient startup, the frequency and phase error between REF and FB are notably large, leading to a substantial injected current into the loop filter.

Additionally, the initial values of the first-stage counters are disparate, resulting in an initially non-constant output from the first-stage comparator. As the PLL progresses towards a phase-locked state, the output stabilizes, and after seven cycles, the lock detector output signal transitions to a high state.

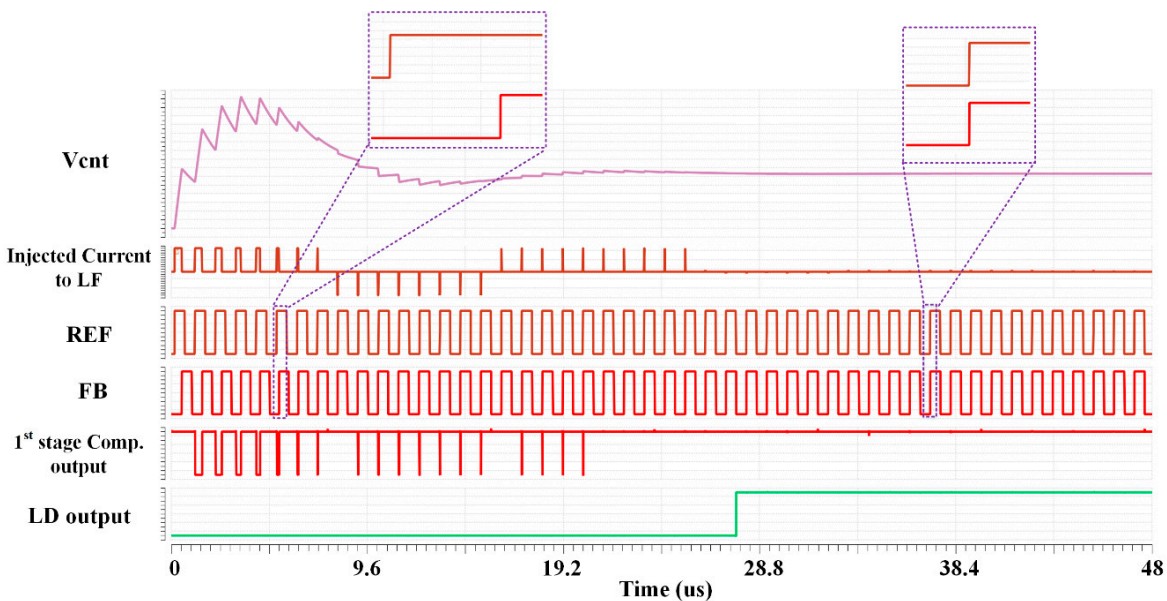

**Figure 9.** Simulated performance of the proposed lock detector.

### 3.4. Low-Power Class-C CMOS LC VCO

A complementary class-C LC VCO topology is chosen to design a low-power VCO at 1.2 GHz (Figure 10a). Class-C harmonic oscillators with large tail capacitors exhibit high conversion efficiency of the DC bias current into class-C current waveforms while minimizing the noise generation, offering an attractive low-power and low-phase noise solution for high-frequency VCOs [25–30]. With a careful design, 3~4 dB phase noise improvement is expected compared to a class-B LC VCO with similar power consumption. The bias voltage, $V_{bias}$, whose level can be adjusted to control the conduction angle, can be generated either on-chip or supplied from an external source. The PMOS cross-coupled pair also operates in class-C mode, since it shares the same bias current with the NMOS cross-coupled pair.

Assuming $I_{ss}$ represents the VCO bias current and $R_P$ is the parallel resistance of the VCO tank ($R_P \approx QL\omega$), we can deduce that the output peak-to-peak swing is approximately $2I_{SS}R_P$ [30]. Setting $I_{SS} = 200$ µA, to achieve a swing$_{out}$ greater than 500 mV implies $R_P > 1$ kΩ. This requirement calls for a high-quality factor (Q) inductor. In this case, $Q \times L_{(nH)} > 160$. The 11-turn inductor exhibits L~20 nH with Q > 13 and the Self-Resonance Frequency (SRF) > 3.3 GHz at 1.2 GHz while occupying 330 µm × 312 µm. The variation of the inductance and Q across the frequency is also provided in Figure 10b. To avoid pushing the cross-coupled pairs into the triode, which reduces the conversion efficiency, the output swing is carefully chosen and limited to the lowest MOS threshold voltage (in this case, Vth, p). This design uses VBias = 600 mV to ensure startup in all process corners.

Fine-tuning of the VCO frequency is done using varactors, while the coarse tuning is accomplished using a three-capacitor array (Figure 10a). The post-layout simulated tuning curves of the VCO in TT corner at T = 27 °C are presented in Figure 11a, covering the entire frequency range needed for BLE operation. Similar tuning range simulations across process, voltage, and temperature (PVT) reveal a relatively small (<7%) deviation from the TT corner simulations (Figure 11b,c). The VCO frequency fluctuations with the supply voltage are particularly small (~4 MHz), indicating acceptable Power Supply Rejection (PSR) performance. The post-layout simulated phase noise of the VCO when biased in

the class-C and class-B regions is shown in Figure 11d. The resulting phase noise is ~2 dB better than the class-B biased VCO at 1 MHz offset.

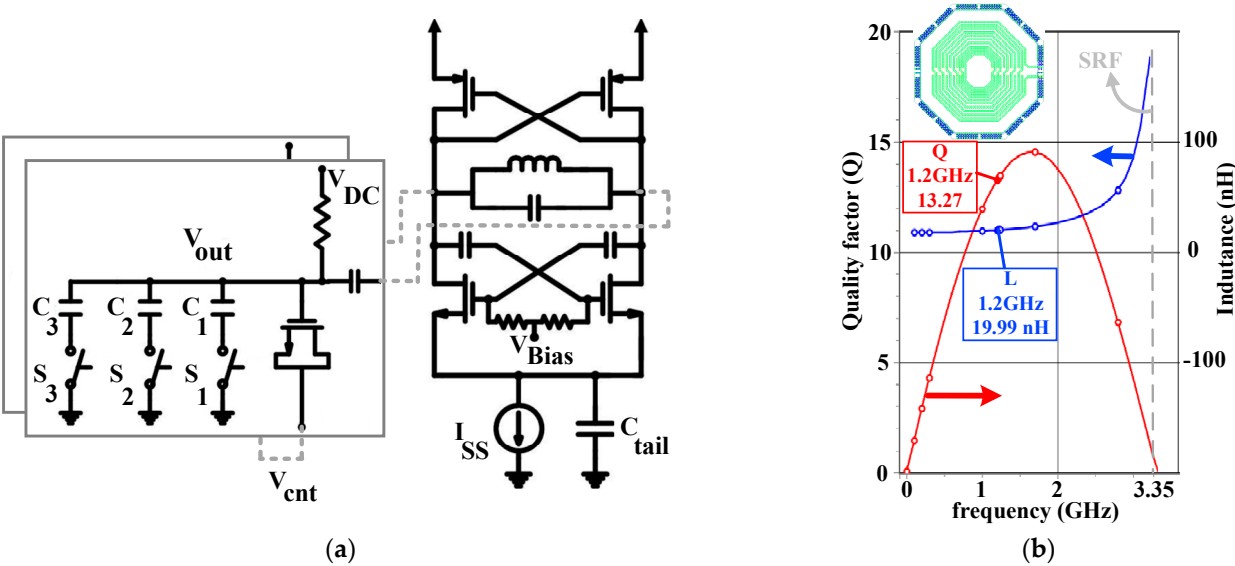

**Figure 10.** (**a**) Class-C LC VCO Schematic; (**b**) simulated performance of the designed inductor (SRF: Self-Resonance Frequency).

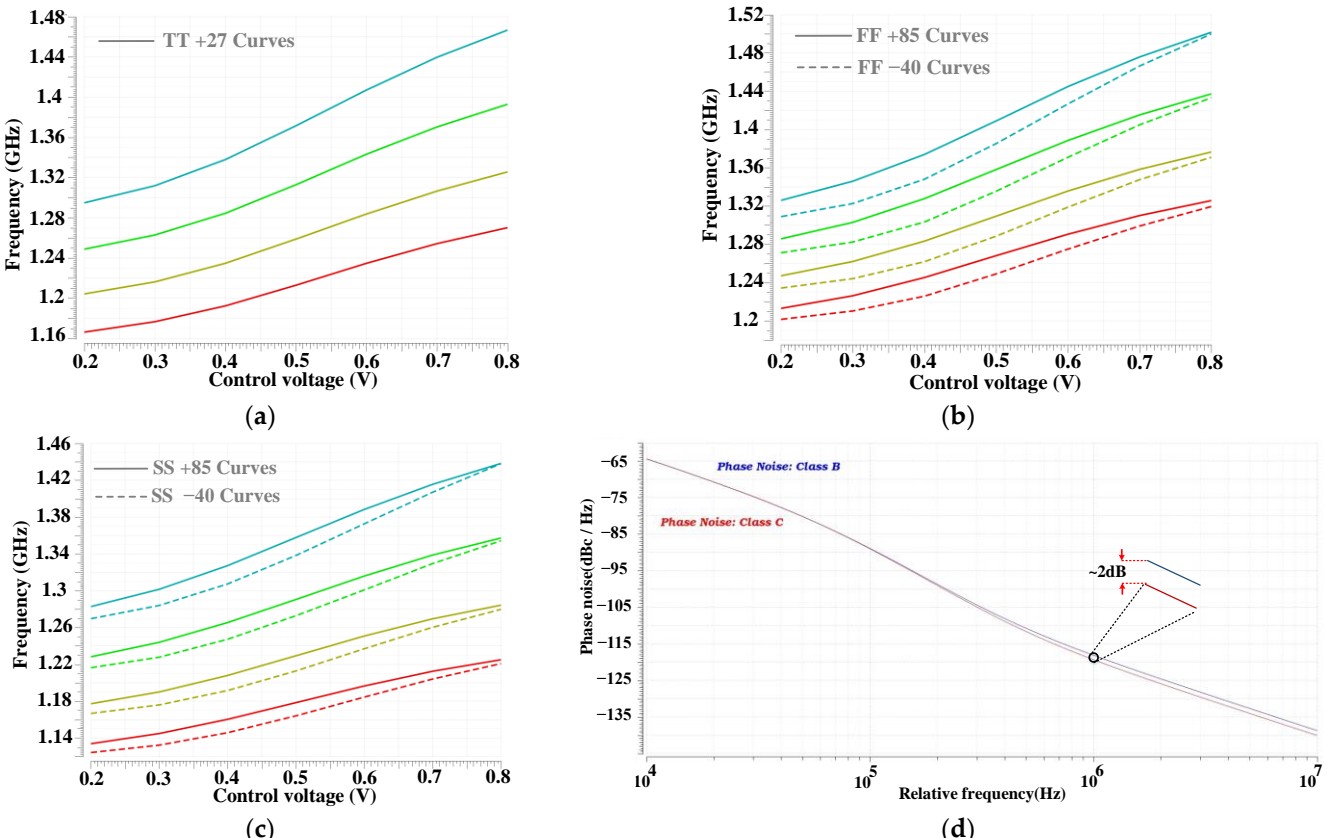

**Figure 11.** Simulated VCO tuning curves (**a**) typical (TT), 27 °C; (**b**) fast (FF), 85 °C and FF, −40 °C; and (**c**) slow (SS), 85 °C and SS, −40 °C; (**d**) VCO simulated phase noise showing both class-B and class-C performance.

Table 1 summarizes the post-layout simulated performance of the class-C VCO, including the Figure Of Merit (FoM) [31] across PVT. As expected, the noise power level is

generally reduced at low temperatures, leading to a better phase noise performance. In slow corners, the threshold voltage increases, particularly for PMOS devices, allowing for larger output amplitude and better phase noise compared to the TT and FF corners at similar temperatures.

**Table 1.** Simulated performance of the class-C LC VCO across PVT.

| Corner & Temp. | Supply (V) | Power Consumption (μW) | Output Swing (mV) | Phase Noise @ 1 MHz (dBc/Hz) | Phase Noise @ 3 MHz (dBc/Hz) | FOM @ 3 MHz (dB) | $K_{VCO}$ (MHz/V) |
|---|---|---|---|---|---|---|---|
| TT, 27 | 1 | 150.1 | 380 | −119.6 | −129.5 | 189.7 | 153 |
| | 1.05 | 152.4 | 383 | −120.4 | −129.6 | 189.8 | 155.1 |
| | 0.95 | 147.5 | 375 | −117.1 | −125 | 185.3 | 144 |
| FF, 85 | 1 | 150.5 | 300 | −117.5 | −127.4 | 187.6 | 129 |
| | 1.05 | 152.9 | 303 | −118.2 | −127.6 | 187.7 | 131 |
| | 0.95 | 148.1 | 296.5 | −114.8 | −123 | 183.3 | 122.5 |
| FF, −40 | 1 | 150.1 | 343 | −122 | −131.8 | 192 | 150 |
| | 1.05 | 152.5 | 346.5 | −122.7 | −131.9 | 192 | 152 |
| | 0.95 | 147.4 | 339 | −119.7 | −127 | 187.3 | 142 |
| SS, 85 | 1 | 151 | 348 | −119.1 | −128.8 | 189 | 155 |
| | 1.05 | 153.4 | 351.5 | −117.3 | −127.3 | 187.5 | 157 |
| | 0.95 | 148.6 | 344 | −115.4 | −126.9 | 187.5 | 146.5 |
| SS, −40 | 1 | 151.5 | 412 | −123 | −131.8 | 192 | 181 |
| | 1.05 | 154 | 416 | −121 | −130.2 | 190.4 | 183.5 |
| | 0.95 | 149 | 407.5 | −119.2 | −129.7 | 190 | 171 |

## 4. INPLL Simulation Results

The proposed INPLL is designed using a 1P9M 65 nm standard CMOS process. The INPLL core (excluding pads) occupies 700 μm × 350 μm of chip area (Figure 12a). The INPLL operates at 1.2 GHz and has a locking range of ~250 MHz (~20% of the carrier frequency). The total power consumption of the INPLL is 350 μW, of which 150 μW is burned in the VCO, 60 μW in the LSPD and Gm, and 140 μW consumed by the remaining blocks (e.g., the dividers, LD, VCO, and REF Buffers).

The simulated settling behavior of the proposed INPLL during startup at TT, 27 °C, is depicted in Figure 12b. For comparison, the settling behavior of an INPLL with similar loop parameters (phase margin and loop bandwidth) that utilizes a conventional PFD-CP architecture is also presented. The conventional INPLL incorporates the same VCO, MMDIV, and LPF (i.e., R = 80 kΩ, C1 = 100 pF, and C2 = 10 pF) as the proposed INPLL. The $V_{cnt}$ ripples at the reference frequency, $f_{REF}$, is decreased from about 700 μV for the conventional INPLL to 30 μV in the LSPD-based INPLL, since the non-idealities associated with the conventional PFD-CP are eliminated. The phase noise performance of the two INPLLs is also simulated and compared with each other (Figure 13a). The noise contribution of critical blocks is also shown in the figure. For the proposed LSPD-based INPLL, the simulated integrated phase noise (IPN) from 10 kHz to 10 MHz is ~−37 dBc, resulting in ~0.02 rad rms jitter (2.9 ps rms jitter at 1.2 GHz). However, the simulated IPN for the conventional INPLL is ~−31.3 dBc, corresponding to ~2× larger jitter, i.e., ~0.038 rad rms or 5.1 ps rms at 1.2 GHz. This jitter reduction is directly attributed to the enhanced phase detection gain achieved through the use of the proposed LSPD. The simulated output spectrum of both INPLLs is shown in Figure 13b. The reference spurs are significantly attenuated, by as much as 22 dBc, in the proposed LSPD-based INPLL compared to the

conventional INPLL. This improvement is mainly due to eliminating inherent non-idealities of the conventional PFD-CP structure (e.g., switching non-idealities).

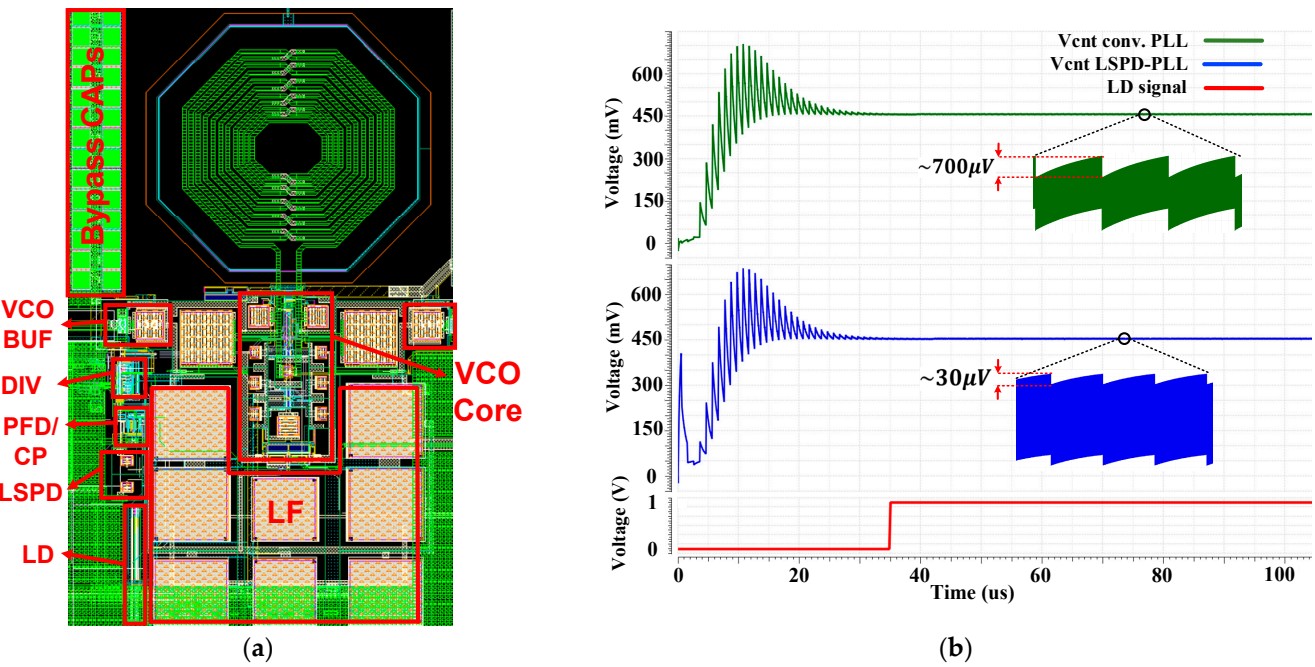

**Figure 12.** (**a**) The layout view of the proposed INPLL; (**b**) simulated settling behavior of the proposed INPLL compared with that of an INPLL that uses conventional PFD-CP.

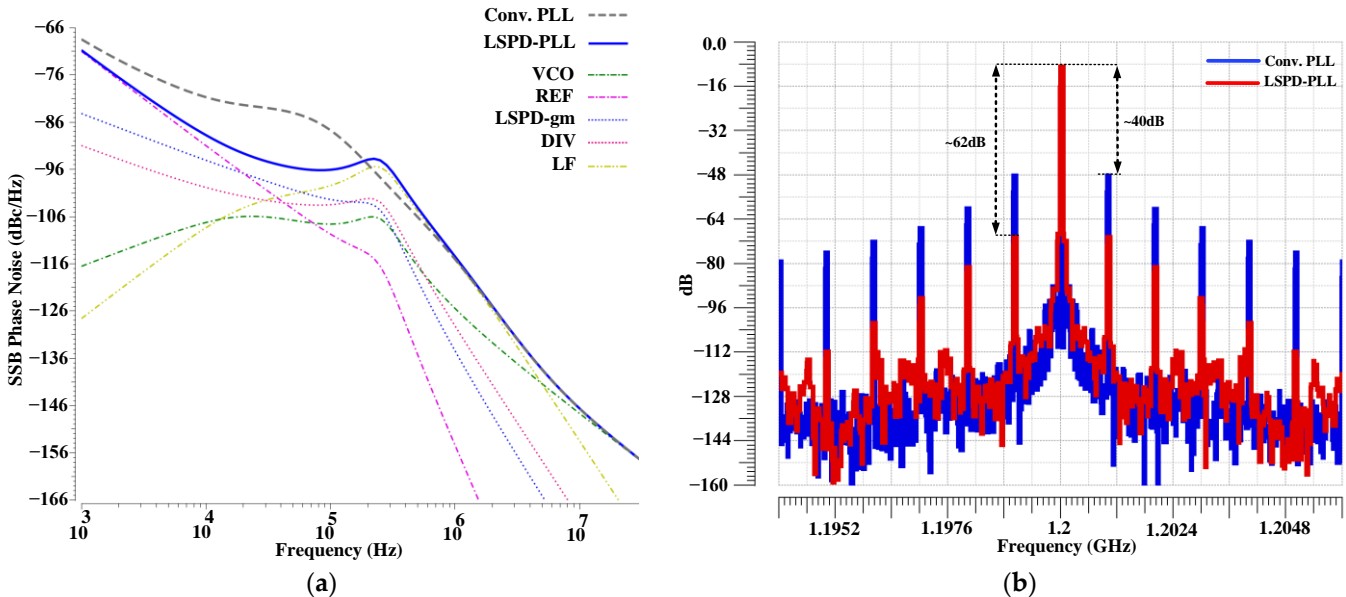

**Figure 13.** (**a**) Simulated phase noise performance of the proposed INPLL compared with that of an INPLL that uses conventional PFD-CP; (**b**) simulated output spectrum of the proposed INPLL compared with that of an INPLL that uses conventional PFD-CP.

The performance of the proposed LSPD-based INPLL is summarized in Table 2 and compared with those of the state-of-the-art INPLLs designed for IoT applications. The proposed INPLL shows competitive performance compared to the state-of-the-art INPLLs while consuming significantly less power. The improved performance in other works is made possible by using high-frequency references to eliminate the degradation caused by a

high up-conversion ratio at the expense of the channel selectivity, i.e., the design cannot cover all BLE channels.

**Table 2.** Proposed INPLL performance comparison with state-of-art 2.4 GHz INPLLs.

|  | **This Work** | **[11]** | **[12]** | **[13]** | **[14]** | **[3]** |
| --- | --- | --- | --- | --- | --- | --- |
| PLL topology (INPLLs) | Analog | Sub-sampling | Analog | Type-I Analog | Sub-sampling | Sub-sampling |
| Phase detection method | LSPD | SSPD | Conv. PFD-CP | XOR-PD & MSSF-LF | SSPD | SSPD |
| Technology | 65 nm | 65 nm | 130 nm | 45 nm | 65 nm | 65 nm |
| Supply voltage (V) | 1 | 0.935 | 1.2 | 1 | 1.2 | 1 |
| Ref. Frequency (MHz) | 1 | 49.15 | 8.66 | 22.6 | 192 | 100 |
| Output frequency (GHz) | 1.18–1.43 | 2.4 | 1.8 | 2.4 | 2.3 | 2.4 |
| Power dissipation ($\mu$W) | 350 | 5860 | 740 | 4000 | 4600 | 900 |
| IPN (dBc) (10 kHz to 10 MHz) | −37 | −44 | −18 | −39.6 | −42 | −55 |
| Rms Jitter (ps) | 2.9 | 0.63 | 15.97 | 0.97 | 0.72 | 0.161 |
| Reference spur (dBc) | −62 | −55.2 | −52 | −65 | −37 | −67 |
| Channel Selection | Y | N | N | N | N | N |
| $FOM_{Jitter}$ [31] | −236.25 | −236.3 | −217.24 | −234.1 | −236 | −256 |

## 5. Conclusions

The paper introduces a low-power 1.2 GHz integer-N PLL designed in a 65 nm standard CMOS process, with a compact core occupying 700 $\mu$m × 350 $\mu$m. Central to its innovation is a novel linear Phase Detector (LSPD), implemented to overcome typical challenges faced by conventional PLL structures operating under low power conditions. The high-gain LSPD showcases a substantial >100× increase in phase detection gain compared to its 1.2 GHz counterpart using the conventional PFD-CP structure. This advancement effectively mitigates inherent issues like poor jitter performance and limited phase detection gain, resulting in a marked reduction in noise contributions from the Phase Detector/Charge Pump (PD/CP), reference clock, and divider paths, consequently yielding significantly lower jitter at the output.

To address the inherent limitations of the LSPD concerning locking range, an adaptive Auxiliary Frequency-Locked Loop (AFLL) is seamlessly integrated. A lock detector accurately identifies the locked state, enabling the strategic disabling of the AFLL to optimize overall jitter performance while conserving power. Further enhancing its capabilities, adopting a complementary class-C LC VCO featuring a high-Q inductor (Q × L > 160), contributes to the design's superior performance. Simulation results demonstrate a notable 2× reduction in jitter, showcasing an equivalent 2.9 ps rms jitter integrated from 10 kHz to 10 MHz (IPN~−37 dBc). Moreover, reference spurs exhibit a substantial 22 dBc improvement compared to conventional INPLLs. Operating at a power-efficient 350 $\mu$W, the proposed INPLL exhibits a decent figure of Merit for jitter ($FoM_{jitter}$) of ~−236 dB, positioning it as a compelling choice for low-power Internet of Things (IoT) applications.

**Author Contributions:** Conceptualization, J.T. and H.M.L.; Investigation, J.T. and H.M.L.; Methodology, J.T. and H.M.L.; Software, J.T.; Supervision, H.M.L. and S.S.; Visualization, J.T.; Writing—original draft, J.T. and H.M.L. and S.S.; Writing—review & editing, J.T. and H.M.L. and S.S. All authors have read and agreed to the published version of the manuscript.

**Funding:** This research received no external funding.

**Data Availability Statement:** All the materials used in the study are mentioned within the article.

**Conflicts of Interest:** The authors declare no conflict of interest.

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
