# Peer review of "An Ultra Low Power Integer-N PLL with a High-Gain Sampling Phase Detector for IOT Applications in 65 nm CMOS"

_jlpea, doi:10.3390/jlpea13040065_

Round 1

Reviewer 1 Report

Comments and Suggestions for Authors

“An Ultra Low Power Integer-N PLL with a High-Gain Sam-2 pling Phase Detector for IOT applications in 65-nm CMOS” is certainly an interesting report on design of low power PLL circuits. While generally nice it has some flaws. Most important is related figures – all the abbreviations and markings on figures need clear and comprehensive descriptions. Unfortunately, it is not true in case of current paper. Also, clarity of the explanatory conceptual structures should be enhanced. In some cases, lettering of subfigures is flawed. Also, designators and abbreviations used in text need thorough explanation. As an example “2ISSRP[12], RP” form page 11 needs clear explanation. In addition, there are two RP’s. Which is correct or are they different? Some minor errors can be detected as well. Like jitter of 2,6 ps in table 2 and 2,9 ps jitter in conclusions and abstract.

Comments on the Quality of English Language

.

Reviewer 2 Report

Comments and Suggestions for Authors

The Authors have presented a novel topology of high-gain phase detector, and used it to design an integer-N PLL. The idea and performance are clearly presented, but comparison to SOA at the end of page 13 needs improvement:

- The Authors compare a simulated result to all measured circuits in the references: do you think to present any measurement?

- the simulated VCO is designed at 1.2 GHz and compared to measured VCOs operating at at least 2.4 GHz : what about the phase noise?

- the references 18-20 are missing

Moreover, at the end of page 3 you could discuss also the effect on the LPF transfer function

Typos:

- the acronym DPE is defined several times

- in several cases (also in figures) the uppercase V has to be used for Volt

- line 165, a space is missing: time(i.e.

- line 202, the voltage differential: the differential voltage

- line 327: With a careful design, 3~4 dB...

- line 336 and Figure 9: RP?

- line 339: 330 μm× 312 μm.

- line 340: variation of the inductance and Q across the frequency are

- line 374: please describe better the design of the conventional PLL

Round 2

Reviewer 1 Report

Comments and Suggestions for Authors

Unfortunately, the issue with the figures has not been entirely resolved, few notes:

-          Very general and applies everywhere - if designators are present on the drawings, then at least quick explanation is in order for ALL of them, without exception: what they are, why they are there. Unfortunately, it is still not the case here. Like Φxyz – what are these?

-          Some elements seem to appear only in text and formulas, like R1, C1, C2 on page 3, and only on page 8 you get to see them. It would be nice to see them immediately.

-          Equations: is it “s” or “S”, and what it is (even if sort of obvious)?

And so on. So it is hard to recommend the paper to immediate publication, even if it is improved.

Comments on the Quality of English Language

.
